# Plasma Electrolysis Spraying Al$_2$O$_3$ Coating onto Quartz Fiber Fabric for Enhanced Thermal Conductivity and Stability

**Aiming Bu, Yongfu Zhang, Yan Xiang, Yunjie Yang, Weiwei Chen \*, Huanwu Cheng and Lu Wang**

Department of Materials Science and Engineering, Beijing Institute of Technology, Beijing 100081, China; buaiming@foxmail.com (A.B.); gyshuishangpiao@163.com (Y.Z.); 15624952172@163.com (Y.X.); yyj163email@163.com (Y.Y.); chenghuanwu@bit.edu.cn (H.C.); wanglu@bit.edu.cn (L.W.)
\* Correspondence: wwchen@bit.edu.cn; Tel.: +86-10-68912709 (ext. 109)

**Abstract:** This manuscript reported the synthesis of Al$_2$O$_3$ coating onto quartz fiber fabric by plasma electrolysis spray for enhanced thermal conductivity and stability. The nano- and micro-sized clusters were partially observed on the coating, while most coating was relatively smooth. It was suggested that the formation of a ceramic coating was followed as the nucleation-growth raw, that is, the formation of the coating clusters was dependent on the fast grow-up partially, implying the inhomogeneous energy distribution in the electrolysis plasma. The deposition of the Al$_2$O$_3$ coating increased the tensile strength from 19.2 to 58.1 MPa. The thermal conductivity of the coated quartz fiber was measured to be 1.17 W m$^{-1}$ K$^{-1}$, increased by ~45% compared to the bare fiber. The formation mechanism of the Al$_2$O$_3$ coating was preliminarily discussed. The thermally conductive quartz fiber with high thermal stability by plasma electrolysis spray will be widely used in flexible thermal shielding and insulation materials.

**Keywords:** quartz fiber; Al$_2$O$_3$ coating; plasma electrolysis spraying; tensile strength; thermal conductivity

## 1. Introduction

Quartz fiber has excellent mechanical properties and high thermal stability, light weight, small thermal expansion coefficient [1–6]. It is an ideal structural material to meet the requirements of high performance and light weight. It has become a commonly used material in the aerospace industry after metal alloys. Quartz fiber was not only light in weight, but also has good load-bearing capacity and maintains high stability [7–9]. In recent years, the outstanding mechanical and dielectric properties of quartz fiber have been used in aerospace high temperature radome system, which need high strength and toughness. However, it usually fails under the action of high load.

Therefore, increasing the tensile resistance of quartz fibers performance was of great importance [10–13]. Meanwhile, the demand for high thermal conductivity materials has been increasing. However, metal materials were susceptible to the environment, resulting in decreased performance, lifetime and substantial loss to the production and living [14–16]. In addition, the density of metal materials is too high to be ignored. Therefore, improving the thermal conductivity of quartz fiber is a hot topic of current research. Coating on quartz fibers is considered to be one of the most effective methods to prevent this in the current situation [17–21]. Wang et al. evaluated the filament tensile strength of the bare and PBO-coated quartz fiber before and after 500 °C treated for 30 min. The flexural strength of PBO-coated fiber-reinforced composites is about 50 MPa [22]. Tao et al. studied the tensile strength of the coated quartz fiber fabrics after calcination at 800 °C for 1 h. The tensile

strength of quartz fiber fabric increased gradually with increasing MoSi2 contents in the coatings [23]. A number of techniques have been deposited in various types coatings onto fiber surface, including chemical vapor deposition, laser ablation, and dip coating. However, these methods have long experimental cycles, and it is still a huge challenge to achieve a uniform thickness coating on the fiber surface [24–26].

We have previously prepared $Al_2O_3$ coating on the continuous quartz fiber fabric by electrode plasma electrolysis in which the samples were not used as electrodes [27]. Based on the nonelectrode plasma electrolysis, we patented a novel technique, i.e., plasma electrolysis spray, in order to rapidly prepare a coating onto large-scaled fiber fabrics [28]. In the present paper, the plasma electrolysis spray was applied to successfully prepare $Al_2O_3$ coatings on quartz fiber fabric. In this technology, plasma was generated by cathode, sprayed on the surface of quartz fiber with electrolyte under the action of thermal compression and mechanical compression, and the $Al_2O_3$ coating was formed by physical and chemical reaction. The tensile properties of the quartz fibers treated for 30 min at 700 °C and then treated for 10 min at 800 °C were studied. The effect of the coating on the high thermal conductivity of quartz fibers was investigated. The formation mechanism of the $Al_2O_3$ coating was preliminarily discussed [29,30]. We believe that the thermally conductive quartz fiber with high thermal stability by plasma electrolysis spray will be widely used in flexible thermal shielding and insulation materials.

## 2. Experimental

### 2.1. Plasma Electrolysis Spraying

Figure 1 shows the schematic diagram of electrolysis plasma spraying $Al_2O_3$ coating onto the quartz fiber fabric. The anode and the cathode were made of graphite and Cu, respectively. The electrolyte was $AlCl_3$ aqueous solution with the concentration of 30 g/L. The electrolyte solution was put in from the tail and out from the nozzle of the spray gun. The voltage was gradually applied between the anode and the cathode until a crucial voltage reached 280 V, where a plasma was formed on the surface of the Cu cathode. The plasma was sprayed out of the nozzle under the effect of the flowing electrolyte with pressure, where the pressure was supplied by a high-pressure pump. The electrolysis plasma provided nucleation and growth-up conditions for the $Al_2O_3$ coating on the quartz fiber fabric. In addition, a large-scaled quartz fiber fabric from Hubei Feilihua Quartz Glass Co., Ltd. and type A plain weave quartz fiber fabric was reasonably coated with the three-dimensional move of the electrolysis plasma spray. In addition, the size of the quartz fiber fabric sample processed in this experiment is 300 × 400 mm. The photo taken during the spray experiment was shown in the inset of Figure 1, where a plasma was obviously observed at the nozzle.

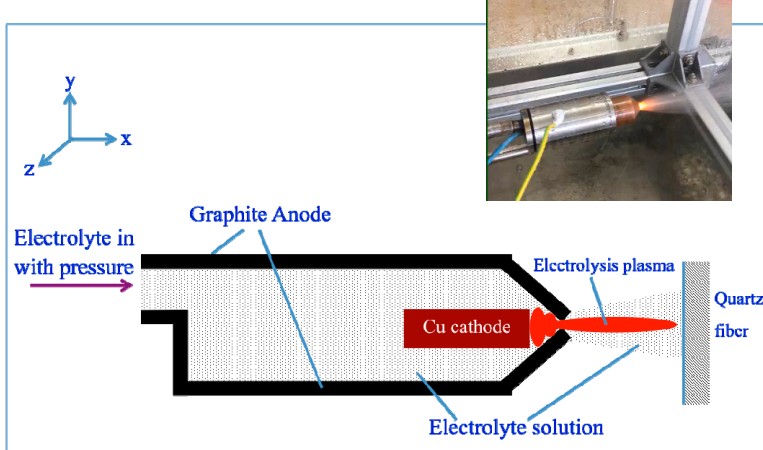

**Figure 1.** The schematic diagram of the experimental process with the spraying photo shown in the inset.

*2.2. Characterization of Coatings*

The microstructures of the bare and the coated quartz fibers were characterized by a scanning electron microscope (SEM, S4800, Hitachi, Tokyo, Japan) and an atomic force microscope (AFM, Bruker Multimode 8). The content of the element in the coating was identified by an energy disperse spectroscope (EDS, S4800, Hitachi). The distribution of the element in the coating was characterized by the EDS mapping mode. The chemical structure of the coating was identified by X-ray photoelectron spectroscopy (XPS, PHI Quantera-II, Tokyo, Japan) and Fourier transform infrared spectroscopy (FTIR, FTIR-8400S_IR-21). All tests are performed at room temperature.

*2.3. Measurement for Properties*

The thermal conductivity was measured by LFA467 Laser Thermal Conductivity Meter at room temperature. In order to precisely measure the thermal conductivity along the longitudinal direction, the fiber bundle was tied into a cylinder with a diameter of 10 mm and then cut into a thickness of 3 mm. Each sample was measured five times in order to obtain the average value. In order to characterize the thermal stability of the quartz fiber, the tensile strength of the quartz fiber was measured after annealing at 700 °C for 30 min and 800 °C for 10 min. The tensile strength was measured by the WDW-E100D Electronic universal testing machine at room temperature. The sample size according to GB type is $\pi(22 \times 2 \text{ cm})$.

## 3. Results and Discussion

*3.1. Surface Morphologies*

Figure 2 shows the surface microstructure of the coating on the quartz fiber. The bare quartz fiber was also characterized for comparison purpose. It is shown from Figure 2a that the bare fiber has a typically smooth surface. The deposition of the coating obviously changed the surface, evidenced by the formation of large amounts of clusters on the surface (Figure 2b). The nano- and micro-sized clusters were partially observed, while most coating was relatively smooth (Figure 2b). It was suggested that the formation of a ceramic coating was followed as the nucleation-growth raw, that is, the formation of the coating clusters was dependent on the fast grow-up partially, implying the inhomogeneous energy distribution in the electrolysis plasma.

Three elements, i.e., Si, O, and Al, existed on the surface of the coated quartz fiber, detected by the EDS result in Figure 2d. Both Si and O were also detected on the fiber substrate (Figure 2c). Obviously, the coating was so thin that the element in the substrate was able to be detected by EDS. In addition, Al was confirmed to result from the coating, with the content up to ~11 wt%. The atom ratio between Si and O was calculated to be 1:1.76 for the coated fiber, compared to 1:1.16 for the bare fiber (Figure 2d). Obviously, the deposition of the coating led to the increase of the O content in the EDS result. Undoubtedly the increased O mainly resulted from the coating, revealing that the coating was mainly composed of Al and O. The distribution of Al and O was relatively uniform in the coating, evidenced by the EDS mapping result in Figure 2e.

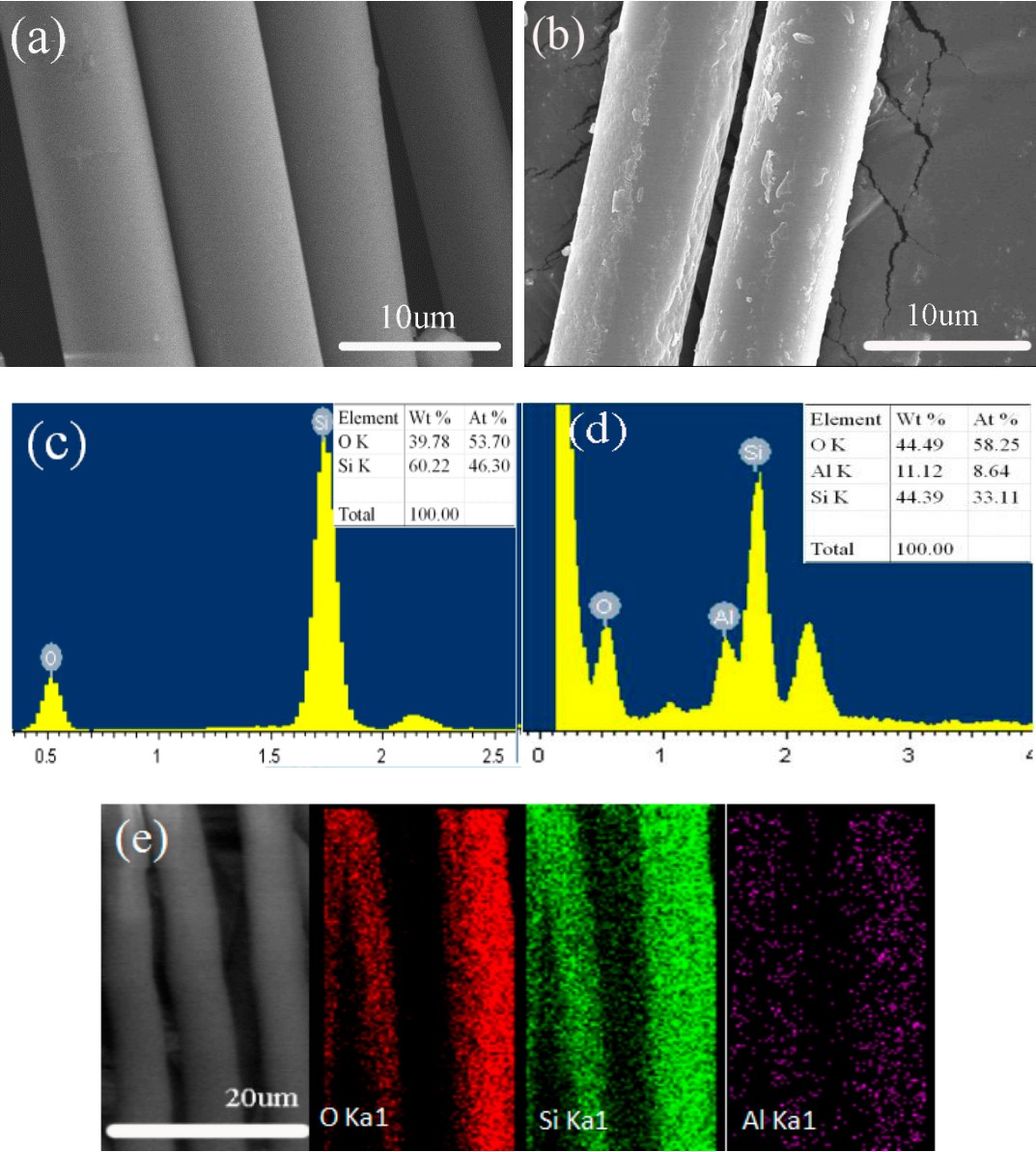

**Figure 2.** SEM morphologies (**a**,**b**), energy disperse spectroscope spectra (**c**,**d**) and mapping results (**e**) of the bare (**a**,**c**) and coated quartz fibers (**b**,**d**,**e**).

### 3.2. XPS and AFM Characterization

The chemical structure of the coating was identified by FTIR and XPS, as shown in Figure 3. The peak position at 780 $cm^{-1}$ was caused by the stretching vibration of Al–O–Al, while the peak of Si–O–Si was located at 1086 $cm^{-1}$ (Figure 3a). Figure 3b shows the XPS result of the uncoated and coated fiber. According to different photoelectron binding energy corresponding peak positions, Al 2p, Si 2p, Al 2s, Si 2s, C 1s, N 1s, O 1s were detected to be present for the coated quartz fiber (Figure 3b). The characteristic binding energy of 73.7 eV corresponded to –Al–O– bond (Figure 3b), consistent with the FTIR result in Figure 3a. Both FTIR and XPS results evidenced the characteristic peak of –Al–O–, corresponding to $Al_2O_3$, consistent with the EDS analysis in Figure 2c,d.

The fine microstructure of the $Al_2O_3$ coating was further characterized by AFM, as shown in Figure 4. A relatively smooth surface was observed on the bare quartz fiber (Figure 4a,b), while the deposition of the $Al_2O_3$ coating increased the surface roughness, evidenced by the formation of the nano-sized $Al_2O_3$ nodules as shown in Figure 4c,d. After the $Al_2O_3$ coating was deposited onto the

surface of the quartz fiber, the root-meansquared roughness of the coated carbon fiber increased from 68 to 136 nm. Therefore, the coating is beneficial to improve the surface roughness of the quartz fiber, and enhance the mechanical interlocking and interface adhesion between the fiber and the matrix in the composite material.

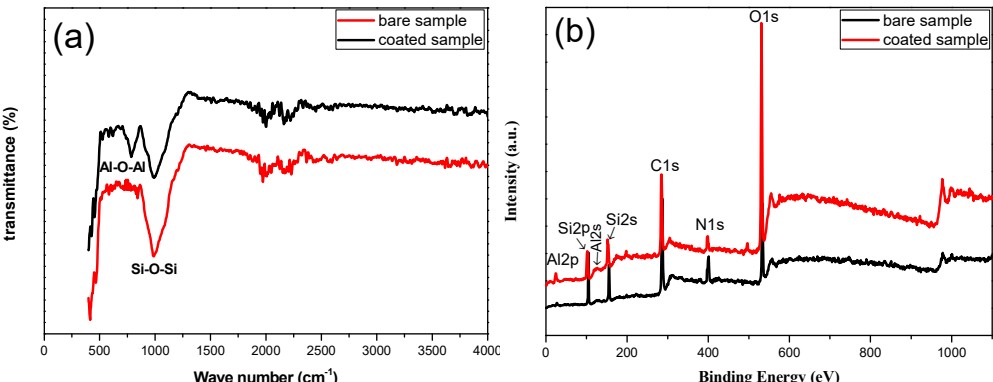

**Figure 3.** FTIR (**a**) and XPS (**b**) spectra of the coated quartz fiber.

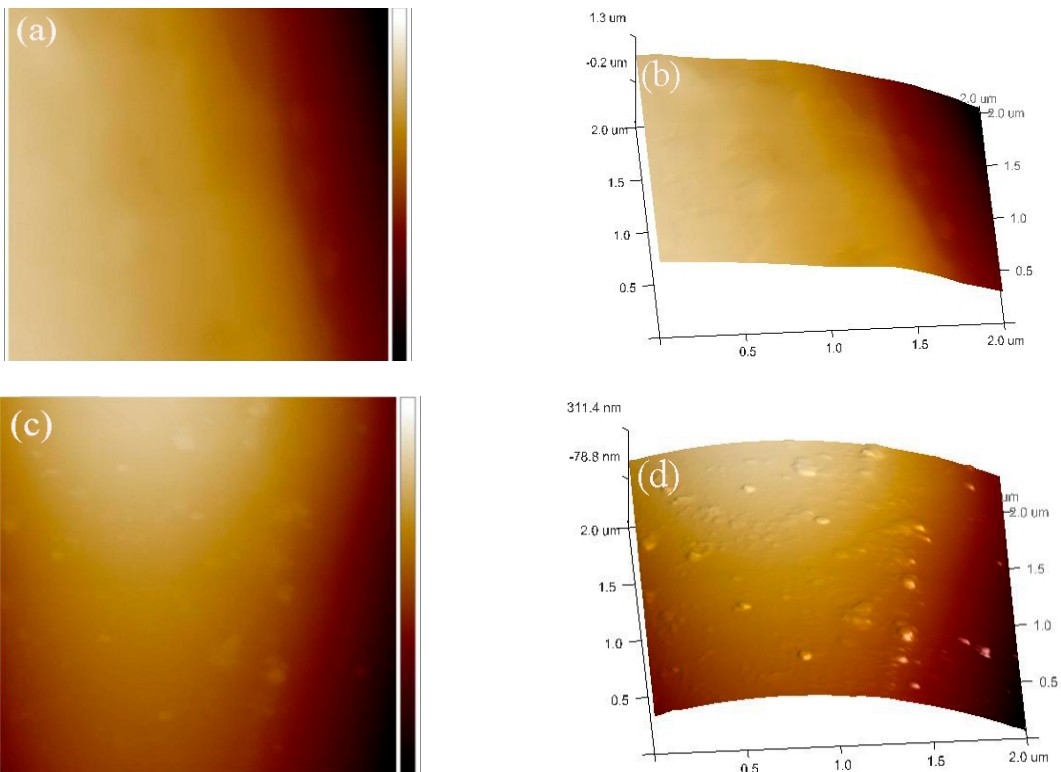

**Figure 4.** AFM surface topographic images of the bare (**a**,**b**) and the coated (**c**,**d**) quartz fibers.

### 3.3. Thermal Conductivity and Stability

Figure 5 shows tensile strength and thermal conductivity of the bare and the coated quartz fibers. The tensile strength of $Al_2O_3$-coated quartz fiber was 58.1 MPa, compared to 19.2 MPa of the bare fiber. Obviously, the tensile strength of quartz fiber was greatly improved by the $Al_2O_3$ coating. The thermal conductivities of coated sample was 1.17 W $m^{-1}$ $K^{-1}$, increased by ~45% compared with the bare sample. The obvious increase of the thermal conductivity mainly resulted from the relatively high conductivity of the $Al_2O_3$ coating. Meanwhile, the $Al_2O_3$ coating also protected the quartz fiber at elevated temperatures.

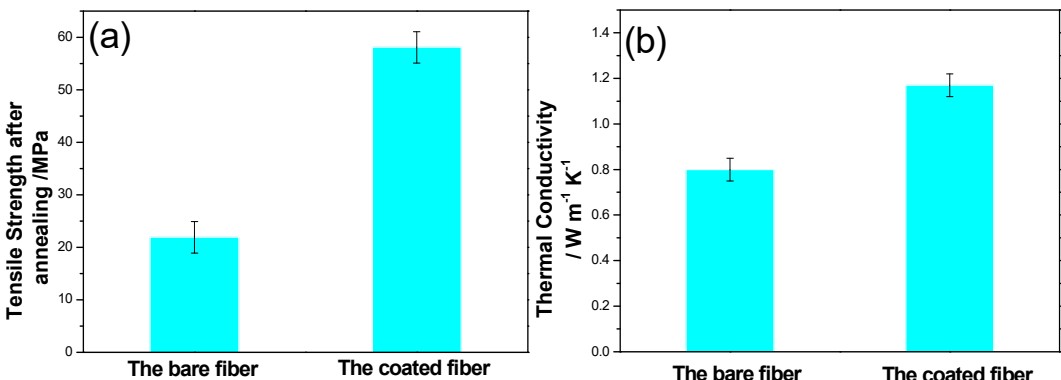

**Figure 5.** The tensile strength (**a**) and the thermal conductivity (**b**) of the bare and the coated quartz fibers.

### 3.4. Formation Mechanism

Based on the above experimental results, it can be clearly concluded that the plasma electrolysis sprayed $Al_2O_3$ coating was able to improve the tensile strength and thermal conductivity of the quartz fiber. The formation mechanism of the $Al_2O_3$ coating has not been fully understood yet. We proposed that the extremely strong potential between the anode and the cathode led to the formation of $Al(OH)_3$ by $AlCl_3$ electrolysis, and the $Al(OH)_3$ gradually evolved into the $Al_2O_3$ under the thermal effect from the plasma arc.

Figure 6 shows the formation mechanism of the $Al_2O_3$ coating on the surface of the quartz fiber fabric. When the working voltage was gradually increased, a hydrogen evolution reaction occurs on the surface of the cathode to release a large amount of $H_2$. When the working voltage reaches a critical voltage, the breakdown of the vapor envelope causes the plasma to discharge. Electrolyte electrolysis forms a large amount of $Al(OH)_3$. As the voltage continues to increase, a stable, uniform, continuous plasma arc was formed, along with the electrolyte ejected from the nozzle. After mechanical compression and thermal compression of the plasma arc, $Al(OH)_3$ dehydrates to form $Al_2O_3$ on the surface of the quartz fiber. The reactions involved are suggested as follows:

Electrolysis of $AlCL_3$ and $H_2O$

$$AlCl_3 \rightarrow Al^{3+} + Cl^- \tag{1}$$

$$H_2O + 2e^- \rightarrow H_2 + 2OH^- \tag{2}$$

Formation of hydroxides

$$Al^{3+} + OH^- \rightarrow Al(OH)_3 \tag{3}$$

$Al_2O_3$ formation

$$Al(OH)_3 \xrightarrow{\text{plasma}} Al_2O_3 + H_2O \tag{4}$$

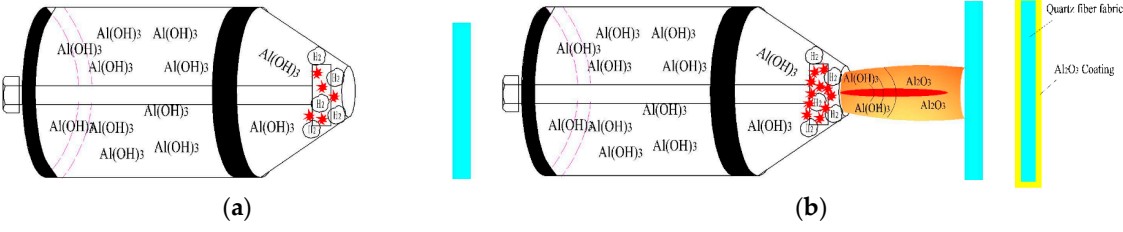

(**a**)            (**b**)

**Figure 6.** The schematic diagram of the formation mechanism of the $Al_2O_3$ coating on quartz fiber fabric, (**a**) $H_2$ bubble and micro arc formation, (**b**) stable plasma jet formation, and $Al_2O_3$ coating deposited on the quartz fiber.

## 4. Conclusions

A thermally conductive quartz fiber with high thermal stability was achieved by plasma electrolysis spraying. The $Al_2O_3$ coating was successfully deposited on the quartz fiber with the $AlCl_3$ aqueous solution. Its tensile properties and high thermal conductivity were studied. The coated sample exhibited very good tensile strength properties after annealing. Compared with the bare sample, the tensile strength increased from 19.2 to 58.1 MPa. The thermal conductivity of the coated quartz fiber also improved due to the high thermal conductivity of the $Al_2O_3$ coating. The thermal conductivity of coated quartz fiber was 1.17 W $m^{-1}$ $K^{-1}$, while the bare sample was 0.81 W $m^{-1}$ $K^{-1}$. The formation mechanism of $Al_2O_3$ coating on quartz fiber surface was preliminarily established.

**Author Contributions:** Conceptualization, W.C. and L.W.; methodology, W.C. and H.C.; software, A.B. and Y.Z.; validation, A.B. and W.C.; formal analysis, A.B. and Y.X.; investigation, A.B. and Y.Y.; resources, A.B. and Y.Z.; data curation, W.C. and L.W.; writing—original draft preparation, A.B.; writing—review and editing, W.C.; visualization, A.B.; supervision, W.C. and L.W.; project administration, W.C.; funding acquisition, W.C. All authors have read and agreed to the published version of the manuscript.

**Funding:** This research was funded by the Basic Research Plan, under grant numbers 201720941052 and 201820941208).

**Conflicts of Interest:** The authors declare no conflict of interest.

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
