# Peer review of "Plasma Electrolysis Spraying Al2O3 Coating onto Quartz Fiber Fabric for Enhanced Thermal Conductivity and Stability"

_applsci, doi:10.3390/app10020702_

Round 1

Reviewer 1 Report

The work deals with the Al2O3 coating of the quartz fiber by plasma electrolysis spraying. The characterizations of the coating were carried out by SEM, AFM, EDS, XPS, FTIR, tensile and thermal conductivity analysis. It was revealed that the tensile strength improved from 19.2 to 58.1 MPa and the thermal conductivity improved from 0.81 to 1.17 W m-1 K-1.

The paper is well structured and procedures are well described, supported by results.

However, a few minor typos correction might be needed, before publishing, as follows:

Line 38: Therefore, instead of Thereofroe

Line 52, Line 125: Al2O3, instead of Al2O3.

Line 63: AlCl3 , instead of AlCl3

Line 65: Spray gun, instead of spray gum.

Line 87: ..direction,, has double comma, would be better to delete one of those.

Line 90-91: 700 ºC and 800 ºC, instead of 700oC and 800oC.

Author Response

Response to Reviewer 1 Comments

Point 1: Line 38: Therefore, instead of Thereofroe

Response 1: We are very sorry for our incorrect typing. We have made correction in the revised manuscript (in red).

Point 2: Line 52, Line 125: Al2O3, instead of Al2O3.

Response 2: We are very sorry for our incorrect typing. We have made correction in the revised manuscript (in red).

Point 3: Line 63: AlCl3 , instead of AlCl3

Response 3: We are very sorry for our incorrect typing. We have made correction in the revised manuscript (in red).

Point 4: Line 65: Spray gun, instead of spray gum.

Response 4: We are very sorry for our incorrect typing. We have made correction in the revised manuscript (in red).

Point 5: Line 87: ..direction,, has double comma, would be better to delete one of those.

Response 5: We are very sorry for our incorrect typing. We have made correction in the revised manuscript (in red).

Point 6: Line 90-91: 700 ºC and 800 ºC, instead of 700oC and 800oC.

Response 6: We are very sorry for our incorrect typing. We have made correction in the revised manuscript (in red).

Reviewer 2 Report

To improve the manuscript, please provide additional information and changes:

- in page 1, line 19, Abstract, reformulate the phrase “The deposition of the Al2O3 coating increased the annealing tensile strength…” because the tensile strength was determined for the annealed samples and is incorrect to claim “annealing tensile strength”;

- in page 1, line 38, the phrase “Thereofroe increasing the tensile resistance of quartz fibers…” correct “Thereofroe”;

- in Introduction specify more clear with quantifiable data the state-of-the-art;

- define the potential applications of Al2O3 coated quartz fibers instead of using a general expression “a wide range of applications in industries”;

- in page 2, lines 65-66, Experimental section, specify the voltage range and the value for "the crucial voltage";

- in page 2, lines 69-70, Experimental section, remove the phrase “In this case, a thermally conductive quartz fiber was achieved with high thermal stability” because it is a comment related to the "Results and discussion" section; Also instead using “a large-scaled quartz fiber” specify the size of the quartz fiber, clarify if the deposition process was carried out on a single fiber or multiple fibers and how many fibers, and specify the provenience of the quartz fibers (supplier, grade) used in experimental works;

- in subsection 2.2. Characterization of coatings, specify the equipment type and analysis conditions for SEM, EDS, AFM, XPS, FTIR used in experimental works;

- in subsection 2.3. Measurement for properties, in the title of 2.3. define the properties that were measured; For the measurement for thermal conductivity with LFA467 Laser Thermal Conductivity Meter specify the density values of the coated and uncoated quartz fiber samples, the type of the reference material used to determine the specific heat capacity of the samples, and the temperature at which the LFA measurements were performed since these data are necessary to calculate the thermal conductivity by laser/light flash method (LFA) (https://www.netzsch-thermal-analysis.com/en/landing-pages/principle-of-the-lfa-method/); Also in "Results" section provide the average values of the thermal diffusivity and standard deviation resulted from five measurements (see page 3, line 89);

- in Fig. 1 revise “Electrolyte in with pressure”;

- in subsection 2.3. specify how the tensile strength was determined (equipment type, measurement conditions, sample size);

- clarify why a DSC analysis was not performed for the determination of the thermal stability of the samples;

- In Fig 2, there are unidentified peaks in the EDS spectra, in Fig. 2 c) at around 2.15 keV, in Fig. 2 d) at around 0.2 keV, 1.1 keV, and 2.2 keV; Please provide their identification; Clarify how a single EDS measurement is relevant to quantify the elements; It is recommended to perform at least three measurements on different areas to obtain a mean value and a standard deviation value;

- in page 3, line 105, in the phrase “the coating was so thin that the element in the substrate was able to be detected by EDS” specify the thickness of the coating;

- FTIR and XPS spectra should be provided comparatively for coated and uncoated quartz fiber samples;

- explain the presence of C 1s, and N 1s detected for the coated quartz fiber (Fig. 3b) since C and N elements were not found in EDS analysis;

- specify the main surface roughness parameters determined by AFM for the coated and uncoated quartz fiber;

- in Fig. 5. The tensile strength (a) and the thermal conductivity (b) of the bare and the coated quartz fibers, it is recommendable to present the average values of the tensile strength and thermal conductivity along with their standard deviation values for the bare and the coated quartz fibers both for as obtained samples and in annealed state to observe the differences in properties; In the present figure, the tensile strength is shown for the annealed samples, whereas the thermal conductivity is shown for the as obtained coated samples and bare quartz fibers;

- clarify how the annealing treatment (700oC for 30 min and 800oC for 10 min) was chosen as an optimum one.

Author Response

Response to Reviewer 2 Comments

Comments and Suggestions for Authors

To improve the manuscript, please provide additional information and changes:

Point 1: in page 1, line 19, Abstract, reformulate the phrase “The deposition of the Al2O3 coating increased the annealing tensile strength…” because the tensile strength was determined for the annealed samples and is incorrect to claim “annealing tensile strength”;

Response 1: We are very sorry for our incorrect writing. It should be “tensile strength” instead of“annealing tensile strength”. We have made correction in the revised manuscript.(in red)

Point 2: in page 1, line 38, the phrase “Thereofroe increasing the tensile resistance of quartz fibers…” correct “Thereofroe”;

Response 2: We are very sorry for our incorrect typing. We have made correction in the revised manuscript (in red).

Point 3: in Introduction specify more clear with quantifiable data the state-of-the-art;

Response 3: We have added several quantifiable data the state-of-the-art in the introduction, as highlighted in Page 9 (in red).

Point 4: define the potential applications of Al2O3 coated quartz fibers instead of using a general expression “a wide range of applications in industries”;

Response 4: During the temperature change, the flexible parts made of quartz fiber fabric can be deformed within limits to ensure its close contact with other parts. Therefore, it can be widely used in flexible thermal shielding and insulating materials. We have added in the article (in red).

Point 5: in page 2, lines 65-66, Experimental section, specify the voltage range and the value for "the crucial voltage";

Response 5 We have specified the experiment value of the voltage in the revised manuscript (in red).

Point 6: in page 2, lines 69-70, Experimental section, remove the phrase “In this case, a thermally conductive quartz fiber was achieved with high thermal stability” because it is a comment related to the "Results and discussion" section; Also instead using “a large-scaled quartz fiber” specify the size of the quartz fiber, clarify if the deposition process was carried out on a single fiber or multiple fibers and how many fibers, and specify the provenience of the quartz fibers (supplier, grade) used in experimental works;

Response 6: According to the reviewer’s suggestion, we have removed the phrase “In this case, a thermally conductive quartz fiber was achieved with high thermal stability”; We are very sorry for our incorrect writing, It should be “a large-scaled quartz fiber fabric” instead of“a large-scaled quartz fiber”, it is not quartz fiber but quartz fiber fabric, and we have modified quartz fiber into quartz fiber fabric in the revised manuscript; we have specified the provenience of the quartz fiber fabric in the revised manuscript (in red).

Point 7: in subsection 2.2. Characterization of coatings, specify the equipment type and analysis conditions for SEM, EDS, AFM, XPS, FTIR used in experimental works;

Response 7: We have specified the equipment type and analysis conditions for SEM, EDS, AFM, XPS, FTIR in the revised manuscript (in red).

Point 8: in subsection 2.3. Measurement for properties, in the title of 2.3. define the properties that were measured; For the measurement for thermal conductivity with LFA467 Laser Thermal Conductivity Meter specify the density values of the coated and uncoated quartz fiber samples, the type of the reference material used to determine the specific heat capacity of the samples, and the temperature at which the LFA measurements were performed since these data are necessary to calculate the thermal conductivity by laser/light flash method (LFA) (https://www.netzsch-thermal-analysis.com/en/landing-pages/principle-of-the-lfa-method/); Also in "Results" section provide the average values of the thermal diffusivity and standard deviation resulted from five measurements (see page 3, line 89);

Response 8: We measure the thermal conductivity at room temperature. Five values are measured for each sample and then averaged to get the Fig. 5(b). Sample density is calculated from mass and volume().

Point 9: in Fig. 1 revise “Electrolyte in with pressure”;

Response 9: The electrolyte enters the spray gun under the pressure of the pump, so that the mechanical and thermal compression of the plasma is ejected from the spray gun. Therefore, “Electrolyte in with pressure”in Fig. 1 is correct expression.

Point 10: in subsection 2.3. specify how the tensile strength was determined (equipment type, measurement conditions, sample size);

Response 10: Thank you for your comments. We have added equipment type, measurement conditions and sample size of tensile strength in subsection 2.3(in red)..

Point 11: clarify why a DSC analysis was not performed for the determination of the thermal stability of the samples;

Response 11: Thank you for your comments. Quartz fiber has high thermal stability. This article mainly studies the tensile properties and thermal conductivity of quartz fiber, so its thermal stability has not been studied by DSC.

Point 12: In Fig 2, there are unidentified peaks in the EDS spectra, in Fig. 2 c) at around 2.15 keV, in Fig. 2 d) at around 0.2 keV, 1.1 keV, and 2.2 keV; Please provide their identification; Clarify how a single EDS measurement is relevant to quantify the elements; It is recommended to perform at least three measurements on different areas to obtain a mean value and a standard deviation value;

Response 12: Quartz fiber is gold-sprayed before EDS, so there will be some spurious peaks. For bare samples, the diffraction peaks of Si are too strong, and there will be coverage for miscellaneous peaks. In the experiment, we choose at least three points for testing. In the article, we chose the most representative data, which is also the closest to the mean value.

Point 13: in page 3, line 105, in the phrase “the coating was so thin that the element in the substrate was able to be detected by EDS” specify the thickness of the coating;

Response 13: The above comment is actually what we are investigating in our lab.  We are now using cross-sectional TEM in order to characterize the thickness of the coating. Such investigation was time-consuming and costly.

Point 14: FTIR and XPS spectra should be provided comparatively for coated and uncoated quartz fiber samples;

Response 14: Thank you for your comments. We have added uncoated sample in FTIR and XPS spectra(in red).

Point 15: explain the presence of C 1s, and N 1s detected for the coated quartz fiber (Fig. 3b) since C and N elements were not found in EDS analysis;

Response 15: Due to the sizing agent on the quartz fiber, it can be detected C 1s, and N 1s. The coating is very thin, so the relative content of C and N in the coating sample is reduced. There is a C peak in the EDS analysis; however, we mainly study whether Al2O3 is formed on the surface of the quartz fiber. Therefore, we do not put the C element into the EDS analysis table to study.

Point 16: specify the main surface roughness parameters determined by AFM for the coated and uncoated quartz fiber;

Response 16: Thank you for your comments. We have specified the root-meansquared roughness in article, as highlighted in Page 3-5(in red).

Point 17: in Fig. 5. The tensile strength (a) and the thermal conductivity (b) of the bare and the coated quartz fibers, it is recommendable to present the average values of the tensile strength and thermal conductivity along with their standard deviation values for the bare and the coated quartz fibers both for as obtained samples and in annealed state to observe the differences in properties; In the present figure, the tensile strength is shown for the annealed samples, whereas the thermal conductivity is shown for the as obtained coated samples and bare quartz fibers;

Response 17: Thank you for your comments. The tensile strength and the thermal conductivity in our article are the average values. Each sample was measured five times. And we have added the standard deviation values in Fig. 5. In the figure shown the bare and coated quartz fibers’s tensile strength after annealing. This is because quartz fiber is used in the radome, and its tensile properties are required to meet the requirements of 700oC for 30 min and 800oC for 10 min.

Point 18: clarify how the annealing treatment (700oC for 30 min and 800oC for 10 min) was chosen as an optimum one.

Response 18: This is because quartz fiber is used in the radome, and its tensile properties are required to meet the requirements of 700oC for 30 min and 800oC for 10 min.

Reviewer 3 Report

Dear Authors,

The manuscript describes synthesis and characterization of quartz fibers coated by aluminum oxide. Authors found that such a process improves tensile strength and thermal conductivity of the fiber. The work is well written however, some remarks should be considered to improve the work quality:

Line 52, please rewrite Al2O3; line 63 AlCl3 with subscript digits. You have written: “The formation mechanism of the Al2O3 coating was preliminarily discussed”. Please provide references to the statement. Line 65 you have wrote “spray gum” maybe spray gun? Please specify the voltage was applied in the process. Please specify models of the following devises you have used in the work: SEM, AFM, EDS system, XPS and FTIR. Line 90 and 91 please rewrite the degrees mark. Please improve resolution of Fig 5 and add an error bars to the results. How many samples did you test? The yield strength and the elongation results of the coated fibre compared to the bare also should be presented in the work. Line 141 and line 185 please rewrite thermal conductivity units using superscript. The thickness of the coating must be presented. Please measure it and add the results. It is not clear why the work was named: “nano-coating”, when do you use this word the meaning that the coating is in nano size, however I didn’t find any related measurement. Please explain.

Author Response

Response to Reviewer 3Comments

Point 1: Line 52, please rewrite Al2O3; line 63 AlCl3 with subscript digits. Response1: We are very sorry for our incorrect typing. We have made correction in the revised manuscript (in red).

Point 2:You have written: “The formation mechanism of the Al2O3 coating was preliminarily discussed”. Please provide references to the statement.

Response2: We have discussed in subsection 3.4.

Point 3:Line 65 you have wrote “spray gum” maybe spray gun? Please specify the voltage was applied in the process. Please specify models of the following devises you have used in the work: SEM, AFM, EDS system, XPS and FTIR. Line 90 and 91 please rewrite the degrees mark. Please improve resolution of Fig 5 and add an error bars to the results. How many samples did you test?

Response3: We are very sorry for our incorrect typing. We have made correction in the revised manuscript. We have specified the equipment type for SEM, EDS, AFM, XPS, FTIR in the revised manuscript (in red). Each sample was measured five times in order to obtain the average value(in red).

Point 4:The yield strength and the elongation results of the coated fibre compared to the bare also should be presented in the work.

Response4:Thank you for your comment. Because this topic is a confidential project related to aerospace, only part of the test data can be given. We’re very sorry for that.

Point 5:Line 141 and line 185 please rewrite thermal conductivity units using superscript.

Response5: We are very sorry for our incorrect typing. We have made correction in the revised manuscript (in red).

Point 6:The thickness of the coating must be presented. Please measure it and add the results. It is not clear why the work was named: “nano-coating”, when do you use this word the meaning that the coating is in nano size, however I didn’t find any related measurement. Please explain. 

Response6: The above comment is actually what we are investigating in our lab.  We are now using cross-sectional TEM in order to characterize the thickness of the coating. Such investigation was time-consuming and costly.

Round 2

Reviewer 2 Report

The authors answered satisfactorily at most questions but additional information and changes are necessary:

- in page 1, line 23, Abstract, delete a “will” from “…will will widely used…” and replace with “…will be widely used”;

- in page 2, line 45, change “lexural strength” with “flexural strength”;

- in page 2, line 63, change “…will widely used in…“ with “…will be widely used in…”;

- in page 2, line 76, referring to “a large-scaled quartz fiber fabric” the authors specified only the supplier without answering to the raised question (Point 6) about the size and grade of the large-scaled quartz fiber fabric used in the experimental works;

- referring to the raised question (Point 8) to specify the density values of the coated and uncoated quartz fiber samples, the type of the reference material used to determine the specific heat capacity of the samples, and the temperature at which the LFA measurements were performed since these data are necessary to calculate the thermal conductivity by laser/light flash method (LFA), the answer is not satisfactory since the authors provided the well-known formula of density even it was not requested and specified the room temperature for the LFA measurements but did not provide the values of density, specific heat capacity, thermal diffusivity, and reference material as they were asked. Moreover, the calculation model should be specified (please see https://www.netzsch-thermal-analysis.com/media/thermal-analysis/brochures/LFA_467_HyperFlash_en_web.pdf)

Author Response

Response to Reviewer 2 Comments

Dear Editor and Reviewers,

On behalf of my co-authors, we thank you very much for giving us an opportunity to revise our manuscript entitled “Plasma electrolysis spraying Al2O3 coating onto quartz fiber fabric for enhanced thermal conductivity and stability”(Manuscript ID: applsci-689624). We also highly appreciate the reviewer’s carefulness, conscientious, and the broad knowledge on the relevant research fields, since he/she has given me a number of beneficial suggestions. We have studied reviewer’s comments carefully and have tried our best to revise our manuscript according to the comments. Attached please find the revised version, which we would like to submit for your kind consideration. The main corrections in the paper and the responses to the reviewer’s comments are marked in red in following responses.

We would like to express our great appreciation to you and reviewer for comments on our paper. Looking forward to hearing from you.

Thank you and best regards.

Yours sincerely,

Aiming Bu

Corresponding author: 
Name: Weiwei Chen
E-mail: [email protected]

Comments and Suggestions for Authors

The authors answered satisfactorily at most questions but additional information and changes are necessary:

Point 1:- in page 1, line 23, Abstract, delete a “will” from “…will will widely used…” and replace with “…will be widely used”;

Response 1: We are very sorry for our incorrect typing and grammar. We have made correction in the revised manuscript (in blue).

Point 2: in page 2, line 45, change “lexural strength” with “flexural strength”;

Response 2:We are very sorry for our incorrect typing. We have made correction in the revised manuscript (in blue).

Point 3: in page 2, line 63, change “…will widely used in…“ with “…will be widely used in…”;

Response 3: We are very sorry for our incorrect grammar. We have made correction in the revised manuscript (in blue).

Point 4: in page 2, line 76, referring to “a large-scaled quartz fiber fabric” the authors specified only the supplier without answering to the raised question (Point 6) about the size and grade of the large-scaled quartz fiber fabric used in the experimental works;

Response 4: The method and device mentioned in this paper are not limited to the size of the treated sample. And the size of the quartz fiber fabric sample processed in this experiment is 300*400mm. The sample used in the experiment is a type A plain weave quartz fiber fabric. We have added in the revised manuscript (in blue).

Point 5: referring to the raised question (Point 8) to specify the density values of the coated and uncoated quartz fiber samples, the type of the reference material used to determine the specific heat capacity of the samples, and the temperature at which the LFA measurements were performed since these data are necessary to calculate the thermal conductivity by laser/light flash method (LFA), the answer is not satisfactory since the authors provided the well-known formula of density even it was not requested and specified the room temperature for the LFA measurements but did not provide the values of density, specific heat capacity, thermal diffusivity, and reference material as they were asked. Moreover, the calculation model should be specified (please see https://www.netzsch-thermal-analysis.com/media/thermal-analysis/brochures/LFA_467_HyperFlash_en_web.pdf)

Response 5: Thank you for your comment. We’re very sorry for not fully understanding your suggestion last time. The thermal conductivity is calculated by the following formula:  

where λ represents the thermal conductivity [W/(m·K)], a is the thermal diffusivity [mm²/s], cp represents the specific heat [J/(g·K)] and ρ is the density [g/cm3].

thermal conductivity [W/(m·K)]

thermal diffusivity [mm²/s]

specific heat [J/(kg·K)]

 density [g/cm3]

Uncoated fiber

0.81

0.68×109

0.54×10-6

2.2

Coated fiber

1.17

0.78×109

0.63×10-6

2.4

Reviewer 3 Report

Dear Authors,

The manuscript was mostly revised, but several issues have been not answered:

Point 2: You have written: “The formation mechanism of the Al2O3 coating was preliminarily discussed”. Please provide references to the statement.

Here you must add a similar works in the field. The discussion you have provided is your own!

Point 4: The resolution of Figure 5 must be improved, otherwise while printing the plot is not clear.

I’m sorry but I don’t understand what does the work confidential is mean. If the work is confidential, please keep all results otherwise, if you have decided to publish them, please provide the full mechanical properties tests results.

Point 5: thermal conductivity units in line 202 must be rewritten again

Point 6: Once again, if you decided to publish an article and you have mentioned that the coating is nano-sized, you must show it. The name of the manuscript has been change, however the meaning is not. Also, I find a lot of meanings of the nano-sized coating through the manuscript.

Author Response

Response to Reviewer 3Comments

Dear Editor and Reviewers,

On behalf of my co-authors, we thank you very much for giving us an opportunity to revise our manuscript entitled “Plasma electrolysis spraying Al2O3 coating onto quartz fiber fabric for enhanced thermal conductivity and stability”(Manuscript ID: applsci-689624). We also highly appreciate the reviewer’s carefulness, conscientious, and the broad knowledge on the relevant research fields, since he/she has given me a number of beneficial suggestions. We have studied reviewer’s comments carefully and have tried our best to revise our manuscript according to the comments. Attached please find the revised version, which we would like to submit for your kind consideration. The main corrections in the paper and the responses to the reviewer’s comments are marked in red in following responses.

We would like to express our great appreciation to you and reviewer for comments on our paper. Looking forward to hearing from you.

Thank you and best regards.

Yours sincerely,

Aiming Bu

Corresponding author: 
Name: Weiwei Chen
E-mail: [email protected]

Comments and Suggestions for Authors

Dear Authors,

The manuscript was mostly revised, but several issues have been not answered:

Point 2: You have written: “The formation mechanism of the Al2O3 coating was preliminarily discussed”. Please provide references to the statement.

Here you must add a similar works in the field. The discussion you have provided is your own!

Response 2: Thank you for your comment. We’re very sorry for not fully understanding your suggestion last time. We have added references in the revised manuscript (in blue).

Point 4: The resolution of Figure 5 must be improved, otherwise while printing the plot is not clear.

I’m sorry but I don’t understand what does the work confidential is mean. If the work is confidential, please keep all results otherwise, if you have decided to publish them, please provide the full mechanical properties tests results.

Response 4: Thank you for your comment. We have improved Figure 5 in the revised manuscript.

We are sorry that we did not provide data on yield stress and elongation. There are two reasons: (â…°) Experiment has a certain confidentiality, and the tensile test is also carried out by a xxx aerospace institute for professional measurement and only provides tensile strength data. (â…±) The quartz fiber fabric was made into a GB type п(22*2cm) tensile sample by being dipped into the resin. It is belongs to brittle fracture, so the yield strength is almost equal to the tensile strength, and the elongation is almost 0.

Point 5: thermal conductivity units in line 202 must be rewritten again

 Response 5: We are very sorry for our incorrect typing. We have made correction in the revised manuscript (in blue).

Point 6: Once again, if you decided to publish an article and you have mentioned that the coating is nano-sized, you must show it. The name of the manuscript has been change, however the meaning is not. Also, I find a lot of meanings of the nano-sized coating through the manuscript.

Response 6: Thank you for your comment. Given that the plasma electrolysis spraying is a novel technique, the present manuscript focuses on the report of the novel method based on the preparation of the coating by the plasma electrolysis spraying on the quartz fiber fabric. The coating has good thermal conductivity and tensile strength. Because the coating is too thin, there is currently no feasible way to determine the thickness of the coating. In this paper, the expression of “nano-coating” is indeed inaccurate. We have revised the full text.

Round 3

Reviewer 2 Report

The authors answered satisfactorily at the raised questions.

Reviewer 3 Report

Dear authors,

Thank you for providing me information.

After the additional revision of the manuscript I think that it is suitable for publication.